# Anti-Oxidant and Anti-Aging Effects of Phlorizin Are Mediated by DAF-16-Induced Stress Response and Autophagy in *Caenorhabditis elegans*

**DOI:** 10.3390/antiox11101996

**Published:** 2022-10-08

**Authors:** Suhyeon Park, Sang-Kyu Park

**Affiliations:** 1Department of Medical Sciences, General Graduate School, Soonchunhyang University, Asan 31538, Korea; 2Department of Medical Biotechnology, Soonchunhyang University, 22 Soonchunhyang-ro, Asan 31538, Korea

**Keywords:** phlorizin, oxidative stress, lifespan, aging, age-related disease, DAF-16, autophagy

## Abstract

Phlorizin (phloridzin) is a polyphenolic phytochemical primarily found in unripe *Malus* (apple). It is a glucoside of phloretin and acts as an inhibitor of renal glucose transport, thus lowering blood glucose. The objective of this study was to determine effects of dietary supplementation with phlorizin on stress response, aging, and age-related diseases using *Caenorhabditis elegans* as a model system. Survival after oxidative stress or ultraviolet irradiation was significantly increased by pre-treatment of phlorizin. Dietary supplementation with phlorizin also significantly extended lifespans without reducing fertility. Age-related decline of muscle function was delayed by supplementation with phlorizin. Phlorizin induced the expression of stress-responsive genes *hsp-16.2* and *sod-3* and nuclear localization of DAF-16, a FOXO transcription factor modulating stress response and lifespan in *C. elegans*. Amyloid-beta-induced toxicity was significantly reduced by phlorizin. This effect was dependent on DAF-16 and SKN-1. Increased mortality induced with a high-glucose diet was partially prevented by phlorizin via SKN-1. Inactivation of dopaminergic neurons observed in a Parkinson’s disease model was completely recovered by supplementation with phlorizin. Genetic analysis suggests that lifespan extension by phlorizin is mediated through oxidative stress response and autophagy. Taken together, these data suggest that phlorizin has strong anti-oxidant and anti-aging activities with potential to be developed as a novel anti-oxidant nutraceutical against aging and age-related diseases.

## 1. Introduction

Aging is one of the most complicated biological pathways observed in all living organisms. Physiological changes occurring during aging include linear decline of the body and bone mass, progressive increase in blood pressure and blood glucose, decreased motility and cardiac output, and so on [1]. At the molecular level, aging can be characterized as the accumulation of DNA mutations, protein oxidation, and lipid peroxidation [2]. However, an underlying mechanism that can explain those age-related changes and increased mortality with time has not been fully understood yet. Many theories of aging have been advanced to elucidate the cause and process of aging. The free radical theory of aging was suggested by Dr. Harman in 1956 and supported by scientific data in various model organisms [3]. It claims that the accumulation of oxidative damage by intracellular or extracellular free radicals is the major causal factor of aging. Old animals show more cellular oxidative damage than young animals [4]. In mammals, maximum lifespans of species are negatively correlated with production levels of reactive oxygen species (ROS), which are the main free radicals found in cells [5]. The telomerase theory of aging suggests that attrition of repeated sequences found at both ends of each chromosome, the telomere, can lead to cellular senescence and organismal aging [6]. The mitochondrial decline theory emphasizes the importance of functional mitochondria for healthy cellular metabolism and bioenergetics. Age-related declines of mitochondrial functions include reduced ATP generation, increased ROS production, decreased number of mitochondria, and changes in mitochondrial permeability [7].

In addition to studies focusing on the understanding of the aging process, people have searched possible nutritional interventions that can retard aging and extend lifespans. The most successful intervention so far is dietary restriction (DR). The effect of DR was first reported in rats [8]. Since then, the longevity phenotype with DR was observed in yeast, nematodes, fruit flies, and mice [9,10]. DR decreased the age-associated increase in oxidative protein modification, DNA mutations, and lipid peroxidation [9]. Dietary-restricted mice and monkeys showed significantly reduced incidence of age-related diseases, including cancer, Alzheimer’s disease (AD), amyotropic lateral sclerosis, and cataracts [11,12]. However, beneficial effects of DR are observed only when DR is maintained continuously, which is hard to apply in human life. Based on the free radical theory of aging, effects of supplementation with anti-oxidants have also been studied using various animal models. Resveratrol, a polyphenolic compound rich in red wine, showed strong anti-oxidant activity and increased lifespan in yeast, nematodes, and fruit flies [13,14]. Intake of resveratrol delayed age-related decline of physical activity and reduced cancer incidence and amyloid beta (Aβ) aggregation [15,16]. Supplementation with vitamin E also extended the lifespan in *Caenorhabditis elegans* and *Drosophila melanogaster* [17]. In mice, a long lifespan was observed only with extremely high dose of vitamins [18]. Chicoric acid, an anti-diabetic caffeoyl derivative, exhibited anti-oxidant activity by scavenging cellular ROS and increased lifespan of *C. elegans* in a dose-dependent manner [19]. Silymarin, a flavanone derivative, increased resistance to oxidative stress, delayed Aβ-induced paralysis, and extended lifespans [20]. Recent studies have reported that cysteine derivatives, N-acetyl-L-cysteine, S-allylcysteine, and selenocysteine, had strong anti-oxidant activity and conferred a longevity phenotype [21,22,23]. Dietary supplementation with coenzyme Q_10_ or lycopene reduced both susceptibility to oxidative stress and tumor incidence. However, it failed to show lifespan extension in mice [24]. 

Phlorizin, also known as phloridzin, is a flavonoid found in the plants of the *Malus genus*. It was first isolated from the bark of apple trees [25]. Flavonoids are phenolic metabolites generated by all plants. They are known to have many biological activities, including anti-oxidant and anti-fungal activities [26]. Phlorizin is a competitive inhibitor of sodium–glucose transporters SGLT1 and SGLT2 [25]. It can reduce plasma glucose concentration by decreasing glucose uptake in the intestine and glucose resorption in kidneys [25]. In a transgenic mouse model of type 2 diabetes mellitus (DM), phlorizin decreased blood glucose levels but increased urine glucose levels [27]. In addition, elevated glycogen synthesis and reduced gluconeogenesis were observed in the liver of type 2 DM mice after treatment with phlorizin [28]. In a type 1 DM model, phlorizin reduced blood glucose levels and body weight, suggesting an anti-obesity activity [29]. Phlorizin also shows anti-tumor and neuroprotective activities. A protective effect of phlorizin has been observed in human cancer cell lines, and cognitive impairment induced by lipopolysaccharide was significantly alleviated by phlorizin treatment [30,31]. Phlorizin has strong anti-oxidant activity. Dietary supplementation with phlorizin can scavenge free radicals, increase activity of anti-oxidant enzymes such as superoxide dismutase (SOD) and catalase (CAT), and prevent lipid peroxidation [32]. 

The aim of the current study was to investigate the anti-aging effect of phlorizin using *C. elegans* as a model system. We examined effects of dietary supplementation with phlorizin on response to environmental stressors, including oxidative stress, heat shock, ultraviolet (UV) irradiation, and lifespan. Changes in physiological and molecular biomarkers of aging, fertility, and motility, and expression of stress-responsive genes in animals treated with phlorizin were monitored. The beneficial effect of phlorizin on age-related diseases was investigated using genetic or nutritional models of AD, DM, and Parkinson’s disease (PD). Possible underlying mechanisms involved in phlorizin-induced longevity were studied using long-lived genetic mutants and genetic knockdown of candidate genes. 

## 2. Materials and Methods

### 2.1. Worm Strains and Maintenance

Wild-type N2 and all transgenic strains were purchased from *C. elegans* Genetics Center (CGC, Minneapolis, USA). CL2070 (dvIs70 [*Phsp-6.2::GFP, rol-6*]) and CF1553 (muIs84 [*Psod-3::GFP, rol-6*]) express GFP with promoter of *hsp-16.2* and *sod-3*, respectively. TJ356 (zls356 IV [*daf-16p::daf-16a/b::GFP, rol-6*]) was used for measuring subcellular localization of DAF-16. CL4176 (dvls27 [*myo-3/Aβ1-42/let UTR, rol-6*]) contains a muscle-specific human Aβ transgene. BZ555 (egls1 [*dat-1p::*GFP]) expresses bright GFP in dopaminergic neurons; *age-1* (U56101) (*hx546*), *clk-1* (U55384) (*e2519*), and *eat-2* (NM064558) (*ad465*) are long-lived mutant strains of *C. elegans*. Worms were grown on Nematode Growth Media (NGM) agar plates (25 mM NaCl, 2.5 mg/mL peptone, 50 mM KPO_4_, 5 μg/mL cholesterol, 1 mM CaCl_2_, 1 mM MgSO_4_, and 1.7% agar) at 20 °C. *Escherichia coli* OP50 was spread on NGM plates as a food source. Phlorizin (Sigma Aldrich, PHL80513) was dissolved and diluted in distilled water.

### 2.2. Resistance to Environmental Stressors

After age synchronization, three-day-old worms were transferred to NGM plates pre-treated with different concentrations of phlorizin (100 μL of each diluted phlorizin solution was spread on NGM plates and dried overnight). After 24 h, individual worms were transferred to single wells of a 96-well plate containing 1 mM hydrogen peroxide (H_2_O_2_) in S-basal medium without cholesterol (5.85 g sodium chloride, 1 g potassium phosphate dibasic, and 6 g potassium phosphate monobasic in 1 L sterilized distilled water) to induce oxidative stress (n = 30). Percent survival was determined after 6 h of incubation. UV irradiation was imposed with 20 J/cm^2^/min of UV for 1 min in a UV crosslinker (BLX-254, VILBER Lourmat Co., Torcy, France). Eight hours of incubation at 35 °C was used for heat shock. Daily survival of worms after heat shock or UV irradiation was recorded until all worms were dead (n = 60).

### 2.3. Intracellular ROS Levels

Age-synchronized worms were grown for seven days after hatching. Twenty worms were randomly selected and transferred to a 96-well black plate containing 195 μL of PBST (single worm in each well) and 5 μL of H_2_DCF-DA (Sigma-Aldrich, St. Louis, MO, USA) was added to each well. After 3 h of incubation, fluorescence intensity was determined in a fluorescence multi-reader (Infinite F200, Tecan, Grodig, Austria).

### 2.4. Lifespan Assay

NGM plates containing OP50 and 12.5 mg/L of 5-Fluoro-2′-deoxyruridine to prevent internal hatching (bagging) were used to monitor lifespans of worms (n = 60). Live and dead worms were counted every day. Live worms were transferred to fresh NGM plates to inhibit starvation. Worms lost, killed, or bagged during the assay were excluded from data analysis. For statistical analysis, the log-rank test was employed [33].

### 2.5. Fertility Assay

Twelve worms were randomly selected at 48 h after hatching and transferred individually to a fresh NGM plate. Each worm was allowed to lay eggs for 24 h and transferred to a fresh NGM plate. This cycle was repeated throughout the gravid period. Eggs spawned for 24 h by individual worms were incubated for additional 48 h at 20 °C. Progeny hatched from eggs were counted daily during the gravid period.

### 2.6. Measurement of Age-Related Decline of Muscle Function

For qualitative analysis of muscle function, locomotive activity of single worms was monitored for 5-, 10-, 15-, and 20-day-old worms (n = 100). It was categorized into three different phases: phase 1, worms that moved spontaneously without mechanical stimuli; phase 2, worms that moved only when mechanical stimuli were given; and phase 3, worms that only moved their heads in response to mechanical stimuli. For quantitative analysis of muscle function, number of thrashings was measured for 10- and 15-day-old worms. Fifteen worms were randomly selected and placed on NGM plates without OP50 individually for 2 min. After adapting in M9 buffer for 10 min, number of thrashings per minute was recorded for each worm.

### 2.7. Expression of Stress-Responsive Genes

Age-synchronized CL2070 and CF1553 worms (n = 20) grown on NGM plates with or without phlorizin were randomly selected and mounted onto glass slides coated with 2% agarose and 1 M sodium azide. Expression of GFP was monitored under a fluorescence microscope and quantified with a fluorescence multi-reader (Infinite F200, Tecan, Grodig, Austria). TJ356 worms were anesthetized with 1 M sodium azide on glass slides at 5, 7, and 9 days after hatching. Nuclear localization of GFP was determined using a fluorescence microscope.

### 2.8. Aβ-Induced Toxicity

CL4176 worms were allowed to lay eggs for 2 h at 15 °C. After removing adult worms, only eggs were incubated for additional 24 h. Sixty hatched worms were selected and incubated at 25 °C for 24 h for induction of human Aβ gene in muscle. After 8 h of incubation, paralyzed worms were counted every hour (n = 60).

### 2.9. High-Glucose-Diet (HGD)-Induced Toxicity

Sixty age-synchronized worms were transferred to NGM plates spread with 100 μL of 40 mM glucose to induce glucose toxicity. Worms were transferred to fresh NGM plates daily during gravid period and every other day after reproduction until all worms were dead. Number of live and dead worms were recorded every day as previously mentioned.

### 2.10. Degeneration of Dopaminergic Neurons

Age-synchronized L3 of BZ555 were treated with 50 mM 6-hydroxydopamine (6-OHDA) and 10 mM ascorbic acid in OP50/NGM solution to induce degeneration of dopaminergic neurons. Solutions were gently mixed every 10 min for 1 h at 20 °C. Worms were washed with M9 buffer three times and transferred onto NGM/OP50 plates containing 10 μM phlorizin and 12.5 mg/L of 5-Fluoro-2-deoxyruridine. After 72 h at 20 °C, worms were mounted onto 2% agarose pads on glass slides with 1M sodium azide. Dopaminergic neurodegeneration was examined using a fluorescence microscope with 485 nm excitation and 530 nm emission filters. The fluorescent intensity in the anterior head region of each worm was measured using image J software. L-3,4-dihydroxyphenylalanie (L-DOPA) was used as a positive control.

### 2.11. Gene Knockdown by RNAi

Ahringer RNAi library was used to obtain RNAi clones for *daf-16* (AF032112), *skn-1* (M84359), and *bec-1* (NM068443) genes [34]. Isopropyl-β-D-thio-galactoside (IPTG, Sigma-Aldrich, St. Louis, MO, USA) was used as an inducer for double-stranded RNA. Cultured RNAi clones of each gene were fed to age-synchronized worms (n = 60). Empty vector (EV) clone was used as a negative control.

### 2.12. Quantitative RT-PCR

Approximately 300 9-day-old worms were collected from 9 cm plates using M9 buffer and washed three times to remove bacteria. Total RNA was extracted using a Trizol reagent (Thermo Fisher Scientific) according to the manufacturer’s protocol and stored at −80 °C. Total RNA (1 μg) was reverse transcribed into cDNA using the ReverTra Ace qPCR RT Master Mix (TOYOBO). Quantitative RT-PCR was performed using 2x SyGreen Mix Hi-ROX (qPCRBIO) according to the manufacturer’s protocol on a StepOne Plus Real-Time PCR System (Applied Biosystems). Expression level of each gene was normalized to the expression of *ama-1* (NM068122). Relative expression levels were calculated using the 2^–ΔΔCt^ method. The primer pairs used for this study are shown in Appendix A.

## 3. Results

### 3.1. Phlorizin Modulated Response to Environmental Stressors

The effect of phlorizin on response to oxidative stress, UV irradiation, and heat shock was measured. Among different concentrations of phlorizin tested, 10 μM of phlorizin significantly increased resistance to oxidative stress. After 6 h of incubation with H_2_O_2_, 76.7 ± 1.92% (mean ± standard error) of worms survived in the untreated control, while 92.2 ± 1.11% of worms survived after being pre-treated with 10 μM of phlorizin (*p* = 0.002) (Figure 1A). There was no significant difference in survival with 1 μM of phlorizin (*p* = 0.651). To determine whether the increased resistance to oxidative stress with 10 μM of phlorizin was due to decreased cellular ROS level, we compared cellular ROS levels in untreated control and worms treated with 10 μM of phlorizin. Cellular ROS levels were not changed by supplementation with phlorizin, suggesting that increased resistance to oxidative stress after treatment with phlorizin was not due to reduced cellular ROS levels (Figure 1B). Survival after UV irradiation was also significantly increased by supplementation with phlorizin. Mean survival time of the untreated control was 4.98 d. It was increased to 5.70 d after treatment with 1 μM of phlorizin (*p* = 0.016), 5.70 d after treatment with 10 μM of phlorizin (*p* = 0.023), and 5.66 d after treatment with 100 μM of phlorizin (*p* = 0.019). There was no significant difference in survival after treatment with 1000 μM of phlorizin (*p* = 0.079) (Figure 1C). However, resistance to heat shock was not affected by supplementation with phlorizin. Survival curves obtained for animals treated with phlorizin were not significantly different from those of untreated control (Figure 1D). Having observed increased survival under oxidative stress and UV irradiation with 10 μM of phlorizin, we decided to use this effective concentration in the following experiments.

### 3.2. Phlorizin Extended Lifespan without Accompanying Reduced Fertility

Based on the free radical theory of aging, we next determined whether phlorizin could confer a longevity phenotype in addition to its anti-oxidant activity. Both mean and maximum lifespan were significantly increased by supplementation with phlorizin. Mean lifespan was extended from 17.5 d for the untreated control to 20.6 d for the phlorizin-treated group (*p* < 0.001, 18.0% increase). Maximum lifespans were 23 d and 27d for the untreated control and phlorizin-treated group, respectively (Figure 2A). Independent replicative experiments also showed a significant increase in lifespan. The mean lifespan was increased from 18.4 d to 20.1 d in the second experiment (*p* = 0.022) and from 18.9 d to 22.2 d in the third experiment (*p* = 0.001) (Appendix A). Previous studies have reported a reduced reproduction in long-lived animals as a trade-off for longevity [35,36]. Total number of progenies produced during the gravid period was not altered by supplementation with phlorizin: 293.0 ± 13.1 in the untreated control and 300.6 ± 7.60 in phlorizin-treated group (*p* = 0.614). As shown in Figure 2B, the number of progenies produced on each day was not different between the untreated control and phlorizin-treated group through the gravid period.

### 3.3. Age-Related Decline of Motility Was Delayed by Supplementation with Phlorizin

To examine the role of phlorizin on age-related muscular dysfunction, we employed both qualitative and quantitative analyses. Locomotive activity was compared between untreated control and phlorizin-treated group at the age of 5, 10, 15, and 20 d. Locomotive activity of each worm was classified into three phases: phase 1, worms that moved spontaneously without mechanical stimuli; phase 2, worms that moved only when mechanical stimuli were given; and phase 3, worms that moved only their head in response to mechanical stimuli. All 5-day-old young animals showed spontaneous locomotive activity with or without supplementation with phlorizin. As expected, locomotive activity declined as worms grew old. In the untreated control, the percentage of worms showing spontaneous locomotive activity with any mechanical stimuli was decreased from 78.0% in 10-day-old worms to 22.0% in 20-day-old worms. This age-related decline of motility was delayed in phlorizin-treated worms. Higher percentages of worms moving spontaneously were observed in the phlorizin-treated group: 92.8% vs. 78.0% (phlorizin-treated group vs. untreated control) in 10-day-old worms, 65.4% vs. 44.3% in 15-day-old worms, and 37.9% vs. 22.0% in 20-day-old worms (Figure 3A). To show the effect of phlorizin on motility quantitatively, we measured the number of thrashings per min. At the age of 10 d, there was a significant increase in the number of thrashings after supplementation with phlorizin. Average numbers of thrashings per min were 99.3 ± 5.28 and 122.5 ± 2.14 in the untreated control and phlorizin-treated group, respectively (*p* = 0.001). Phlorizin increased the average number of thrashings per min from 47.6 ± 6.06 in the untreated control to 68.9 ± 6.42 in the phlorizin-treated group at 15 d after hatching (*p* = 0.046) (Figure 3B). 

### 3.4. Phlorizin Induced Expression of Hsp-16.2 and Sod-3 and Nuclear Localization of DAF-16

Previous studies have found that expression levels of *hsp-16.2* (NM001392482) and *sod-3* (NM078363) are positively correlated with an individual’s lifespan and that they can be used as longevity-assuring biomarkers [37,38]. Supplementation with phlorizin significantly up-regulated expression of both *hsp-16.2* and *sod-3* (Figure 4A). The expression of *hsp-16.2* was increased 1.4-fold in 7-day-old worms and 4.0-fold in 9-day-old worms after treatment with phlorizin compared to the untreated control (both *p* < 0.001). Expression of *sod-3* was also markedly induced in the phlorizin-treated group (2.8- and 3.3-fold increases in 7- and 9-day-old worms, respectively, both *p* < 0.001) (Figure 4B). DAF-16 is a FOXO transcription factor that can regulate the expression of many stress-responsive and age-related genes including *hsp-16.2* and *sod-3* after nuclear localization [39]. Using GFP fused to DAF-16, we determined the subcellular distribution of DAF-16 (Figure 4C). Enhanced nuclear localization of DAF-16 was observed after phlorizin supplementation. The percent of worms showing nuclear localization of DAF-16 were 55.8 ± 1.60% in the untreated control and 64.6 ± 2.92% in the phlorizin-treated worms (*p* = 0.039) in 9-day-old worms (Figure 4D).

### 3.5. Positive Impact of Phlorizin Was Observed in Age-Related Disease Models

Using a genetic or nutritional disease model, we tested effects of phlorizin on age-related diseases. Dietary supplementation with phlorizin significantly delayed Aβ-induced toxicity. The time when 50% of worms paralyzed after the induction of the Aβ transgene in muscle was extended up to 13.5% by phlorizin treatment: 7.2 h in the untreated control and 8.2 h in the phlorizin-treated group (*p* = 0.013) (Figure 5A). Recent studies have found that DAF-16 and SKN-1 are involved in Aβ-induced toxicity [40,41]. Genetic knockdown of full-length *daf-16* or *skn-1* completely abolished delayed paralysis by phlorizin (Figure 5B). This suggests that the inhibitory effect of phlorizin on Aβ-induced toxicity requires DAF-16 and SKN-1 (Figure 5B). Repeated experiments showed similar results (Appendix A). HGD was developed as a nutritional disease model of DM in *C. elegans* [42]. Increased mortality caused by HGD was markedly recovered by supplementation with phlorizin. Mean lifespan was decreased from 18.8 d in the untreated control to 12.9 d in the group fed HGD (*p* < 0.001). However, simultaneous treatment with HGD and phlorizin prevented the toxic effect of HGD (mean lifespan of 21.0 d, *p* < 0.001 compared to worms treated with HGD only) (Figure 5C). RNAi of *skn-1* eliminated the effect of phlorizin on HGD-induced toxicity (Figure 5D). This indicates that SKN-1 is necessary for the prevention of reduced survival under HGD by phlorizin (Figure 5D). Independent replication exhibited a significant inhibition of HGD-induced toxicity by phlorizin and the requirement of SKN-1 for the effect of phlorizin (Appendix A). Degeneration of dopaminergic neurons, a key physiological change observed in PD, was induced by treatment with 6-OHDA but disappeared after simultaneous treatment with 6-OHDA and L-DOPA, a widely used drug for PD. Interestingly, supplementation with phlorizin also markedly prevented degeneration of dopaminergic neurons caused by 6-OHDA (Figure 6A). Treatment with 6-OHDA decreased GFP expressed in dopaminergic neurons to 60.4 ± 5.28% compared to the untreated control (100.0 ± 5.76%, *p* < 0.001). Relative fluorescence was recovered up to 113.9 ± 7.20% by L-DOPA (*p* < 0.001) and 102.9 ± 5.94% by phlorizin (*p* < 0.001) (Figure 6B). Replicative experiments also showed a significant inhibitory effect of phlorizin on dopaminergic neuronal degeneration (Appendix A). 

### 3.6. Lifespan Extension by Phlorizin Was Mediated through Oxidative Stress Response and Autophagy

To identify the underlying mechanisms involved in phlorizin-induced longevity, three mutants representing each known genetic pathway leading to lifespan extension were employed. The lifespan of *age-1*, a long-lived mutant due to reduced insulin/IGF-1-like signaling, was not changed after treatment with phlorizin. Mean lifespan of *age-1* was 29.7 d in the untreated control and 30.7 d in the group of *age-1* treated with phlorizin (*p* = 0.254) (Figure 7A). In the *clk-1* mutant, in which ROS production was reduced by dysfunctional mitochondria to induce a long lifespan, phlorizin failed to increase lifespans. Mean lifespans were 19.5 and 18.7 d for untreated control and phlorizin-treated groups of *clk-1*, respectively (*p* = 0.163) (Figure 7B). The *eat-2* mutation is a genetic model of DR. DR retards aging and age-related physiological changes in many model organisms [9,10]. The mean lifespan of *eat-2* (20.4 d) was not affected by phlorizin treatment (18.6 d, *p* = 0.898) (Figure 7C). Taken together, no additional lifespan extension of long-lived mutants suggests that the lifespan-extending effect of phlorizin overlaps with that of each genetic pathway. We then examined the role of specific factors common to those lifespan-extending pathways. The phlorizin-induced longevity observed in the wild-type N2 completely disappeared when the expression of *daf-16* was repressed. Mean lifespans of N2 with *daf-16* RNAi (12.9 d) and N2 with *daf-16* RNAi and phlorizin treatment (12.8 d) were not significantly different (*p* = 0.941) (Figure 7D). The effect of lifespan extension by phlorizin was also abolished by genetic knockdown of *bec-1*, a worm homolog of a human autophagic gene. Mean lifespans were 18.9 and 18.5 d in N2 with *bec-1* RNAi and N2 with *bec-1* RNAi and phlorizin treatment, respectively (*p* = 0.724) (Figure 7D and Appendix A). Since DAF-16 modulates the response to oxidative stress and BEC-1 is a key factor regulating autophagy, we analyzed the effect of phlorizin on expression of genes involved in oxidative stress response and autophagy. Among four oxidative stress-responsive genes tested, expression levels of *ctl-1* and *sod-3* as downstream targets of DAF-16 were significantly increased after supplementation with phlorizin (1.4- and 1.5-fold increase in mRNA levels of *ctl-1* (*p* = 0.016) and *sod-3* (*p* = 0.027), respectively) (Figure 8). Expression levels of *skn-1* and *gst-4* (NM069447) were also increased after supplementation with phlorizin, although such increases were not statistically significant (*p* > 0.05). The increase in the expression of *bec-1* by phlorizin (1.3-fold) was not statistically significant (*p* = 0.178). The mRNA level of *lgg-1* (NM062876), a well-known autophagic gene and a positive regulator of autophagosome assembly, was induced 1.3-fold by supplementation with phlorizin compared to that in the untreated control (*p* = 0.052) (Figure 8).

## 4. Discussion

Most phytochemicals are secondary metabolites produced in diverse plants to protect against environmental stresses and microbial infections. Phytochemicals have been found in various fruits, vegetables, cereals, and nuts. They can be categorized into several classes according to their chemical structures: phenolic compounds, terpenes, betalains, polysulfides, organosulfur compounds, and so on [43]. Lifespan-extending effects of supplementation with phytochemicals have been reported in various model organisms. Epicatechin, a phytochemical abundant in cocoa bean, increased the lifespan of *C. elegans*, *Drosophila melanogaster*, and mice [44]. In humans, dietary intake of chocolate improved average life expectancy up to 4 years [45]. Butein, a phenolic compound, showed lifespan-extending activity in the yeast *Saccharomyces cerevisiae* [13]. Supplementation with other phytochemicals such as fisetin, kaempferol, and myricetin conferred a longevity phenotype via DAF-16, a FOXO transcription factor regulating expression of anti-oxidant genes in *C. elegans* [43]. We showed that phlorizin, a phenolic phytochemical abundant in apples, could significantly increase both mean and maximum lifespans of *C. elegans*. Age-related decline of motility was also significantly delayed by phlorizin. The lifespan-extending effect of phlorizin has been previously reported in *S. cerevisiae* and *D. melanogaster* [43]. Reduced locomotor function observed in aged animals was partially reversed by phlorizin in fruit flies [46]. Many phytochemicals exhibit anti-oxidant activity through scavenging ROS directly or modulating the expression of anti-oxidant genes. Generally, phytochemicals having hydroxy groups have ROS scavenging activity. However, the exact structure–activity relationship for ROS scavenging activity is still elusive. Quercetin and kaempferol are known to be strong ROS scavengers [44]. The anti-oxidant activity of luteolin, luteolin tetramethylether, and kaempferol, all of which have four hydroxy groups, are markedly different, although all of them have four hydroxy groups [47]. Previous studies suggested that the in vivo anti-oxidant activity of luteolin, curcumin, and resveratrol is due to the activation of stress-responsive transcription factor Nrf2, and not related to direct ROS scavenging activity because of a poor bioavailability in vivo [44]. Nuclear import of DAF-16 was induced by supplementation with myricetin, kaempferol, and quercetin, and activities of its downstream target anti-oxidant genes were also increased [43]. In the present study, supplementation with phlorizin significantly increased resistance to oxidative stress without affecting cellular ROS levels. However, increased nuclear localization of DAF-16 and induction of *hsp-16.2* and *sod-3* as downstream targets of DAF-16 were observed. These findings suggest that the anti-oxidant activity of phlorizin might not be due to its ROS scavenging activity, but due to modulation of DAF-16-mediated anti-oxidant response. Interestingly, *hsp-16.2* and *sod-3* were identified as longevity-assurance genes, as their expression levels were positively correlated with the lifespan of *C. elegans* [37,38]. Further studies focusing on the identification of genetic pathways mediating anti-oxidant and anti-aging activities of phlorizin and the effects on lifespan of the mammalian system will provide a deeper scientific understanding of the bioactivities of phlorizin.

Phytochemicals also show preventive or therapeutic effects on several chronic diseases such as cancer, cardiovascular disease, obesity, and diabetes. Resveratrol reduced oxidation of low-density lipoprotein, which is one of major causes of coronary heart disease and induced cancer cell death [15,48]. Epigallocatechin gallate inhibited the accumulation of Aβ in *C. elegans* [49]. We showed that dietary supplementation with phlorizin exhibited beneficial effects in disease models of AD, DM, and PD. A previous study has shown that phlorizin reduced neuronal damage and improved learning and memory abilities in D-galactose-induced aged mice [50]. In streptozotocin-induced dementia of AD, phlorizin alleviated cognitive decline and neuropathological conditions with AD [51]. Oral administration of phlorizin reversed decreased dopamine, neuroinflammation, and motor dysfunction observed in PD model mice [52]. Recent studies have identified transcription factors that can mediate protective effects of anti-oxidants against AD in *C. elegans*. Fluoxetine significantly reduced Aβ-induced toxicity via DAF-16 [53]. The inhibitory effect of *Terminalia chebula* Retz and otophylloside B on Aβ aggregation was dependent on DAF-16, but not on SKN-1 [54,55]. In contrast, delayed Aβ-induced paralysis by rose essential oil required SKN-1, but not DAF-16 [56]. According to the result presented here, the protective effect of phlorizin against Aβ-induced toxicity was dependent on both DAF-16 and SKN-1. HGD can impair immune response and increase mortality in *C. elegans*. Molecular pathways causing those detrimental effects are mediated by SKN-1 [42]. The recovery effect of phlorizin on the HGD-induced shortened lifespan completely disappeared by genetic knockdown of *skn-1*. This indicates that SKN-1 is required for the prevention of HGD-induced toxicity. Taken together, it is suggestive that phlorizin is a strong phytochemical candidate for the development of nutraceuticals targeting age-related diseases.

Several lifespan-extending pathways have been identified in *C. elegans*. They are well-conserved in other species. The first genetic mutant showing a longevity phenotype was the *age-1* mutant, which could increase the lifespan via reduced insulin/IGF-1-like signaling [57]. Other mutations in genes such as *daf-2* and *daf-16* involved in insulin/IGF-1-like signaling could also regulate the lifespan of *C. elegans* [58]. The *clk-1* mutant produces less ROS due to decreased function of the mitochondrial electron transport chain reaction, leading to increased lifespan [59]. The *eat-2* mutant is a widely used genetic model of DR, since it intakes less food owing to reduced pumping rate in pharynx [36]. Dietary supplementation with phlorizin in three mutants representing each lifespan-extending pathway did not further increase their lifespans, although a significant extension was observed for wild-type N2. These findings suggest that the effect of phlorizin on lifespan overlaps with effects of three genetic pathways. Such effect of phlorizin is mediated by common downstream mechanisms. DAF-16 can bind and activate promoters of many anti-oxidant genes. It is known to be a key downstream effector for lifespan extension via reduced insulin/IGF-1-like signaling and DR [39]. We observed enhanced nuclear location of DAF-16 and increased expression of *ctl-1* and *sod-3* (anti-oxidant genes regulated by DAF-16) in long-lived animals treated with phlorizin. In addition, repression of *daf-16* by RNAi abolished lifespan extension induced by supplementation with phlorizin. Based on these data, we can conclude that improved response to oxidative stress by DAF-16 is one of the major underlying mechanisms involved in phlorizin-induced longevity. Autophagy is a eukaryotic process for degrading and recycling cellular organelles and components during development or starvation. Recent studies have shown that autophagy is increased in many lifespan-extending interventions. Accumulation of vacuolar structures and early autophagosomes was increased in long-lived animals having reduced insulin/IGF-1-like signaling [60]. Autophagy was also triggered by DR in *C. elegans* [61]. Genetic knockdown of autophagic genes, including *bec-1*, *atg-18*, and *unc-51*, eliminated the longevity phenotype observed in *age-1*, *clk-1*, and *eat-2* [62]. This suggests that autophagy is a central mechanism common to known lifespan-extending pathways. Interestingly, the long lifespan conferred by phlorizin treatment disappeared when the expression of *bec-1* was inhibited. Significant induction of *lgg-1*, a gene responsible for autophagosome assembly, was also observed in long-lived worms supplemented with phlorizin. These results suggest that increased autophagy is another major underlying mechanism involved in the lifespan extension caused by phlorizin. The effects of phlorizin on autophagy are necessary to solidate our findings. Since it is very hard to predict the physiological relevance of findings in lower organisms to higher organisms, effects of dietary intervention with phlorizin should be studied in mammals in the near future. Further studies that identify other intracellular changes associated with phlorizin-induced longevity will help us understand the role of phlorizin in aging to provide scientific rationales for the development of novel anti-aging nutraceuticals using phlorizin.

## 5. Conclusions

Dietary supplementation with phlorizin increased resistance to oxidative stress and UV irradiation. Phlorizin also extended lifespans and delayed the age-related decline of motility. Beneficial effects of phlorizin were observed in disease models of AD, DM, and PD. Genetic analysis with mutant strains and RNAi knockdown suggested that the anti-oxidant activity of phlorizin is mediated via DAF-16-dependent stress response, and that the lifespan-extending effect of phlorizin involves DAF-16 and autophagy. In conclusion, phlorizin has strong anti-oxidant and anti-aging activities and can ameliorate the conditions of several age-related diseases.

## Figures and Tables

**Figure 1 antioxidants-11-01996-f001:**
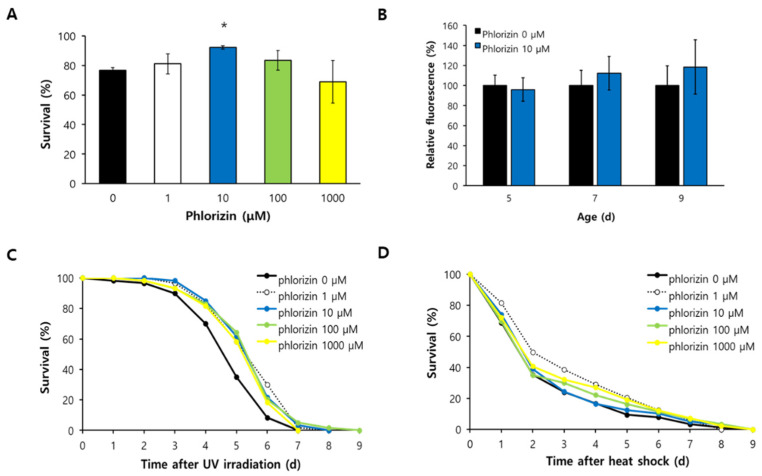
Effects of phlorizin on stress response and cellular ROS levels. (**A**) Resistance to oxidative stress induced by H_2_O_2_ was examined. (**B**) Cellular ROS levels were compared between untreated control and worms treated with 10 μM of phlorizin in 5-, 7-, and 9-day-old worms. Survival after UV irradiation (**C**) or heat shock (**D**) was examined with different concentrations of phlorizin; 0, 1, 10, 100, and 1000 μM. Error bar indicates standard error. *, a significant difference (*p* < 0.05) compared to untreated control (0 μM of phlorizin).

**Figure 2 antioxidants-11-01996-f002:**
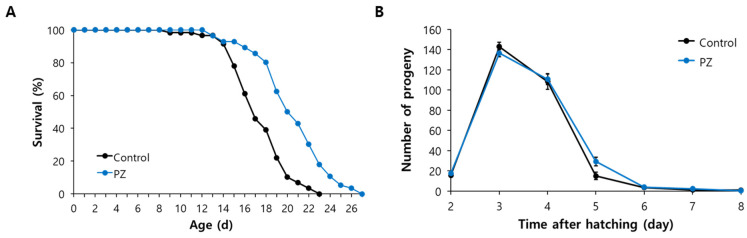
Effects of phlorizin on lifespan and fertility of C. elegans. (**A**) Lifespan assay was performed with sixty age-synchronized worms. Live and dead worms were counted daily and killed/lost/bagged worms were excluded from data analysis. (**B**) Time-course distributions of progeny number produced were compared between untreated control and worms treated with phlorizin during a gravid period (n = 12). Error bar indicates standard error. PZ, phlorizin (10 μM).

**Figure 3 antioxidants-11-01996-f003:**
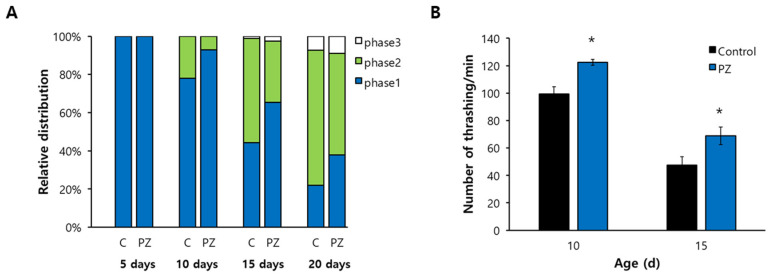
Delayed muscular dysfunction with age by supplementation with phlorizin. (**A**) Locomotive activity was classified into three phases according to motility of each worm (n = 100). (**B**) Number of thrashings was counted for 1 min for 10- and 15-day-old worms under microscope (n = 15). Error bar indicates standard error. C, untreated control; PZ, phlorizin (10 μM); *, a significant difference (*p* < 0.05) compared to untreated control.

**Figure 4 antioxidants-11-01996-f004:**
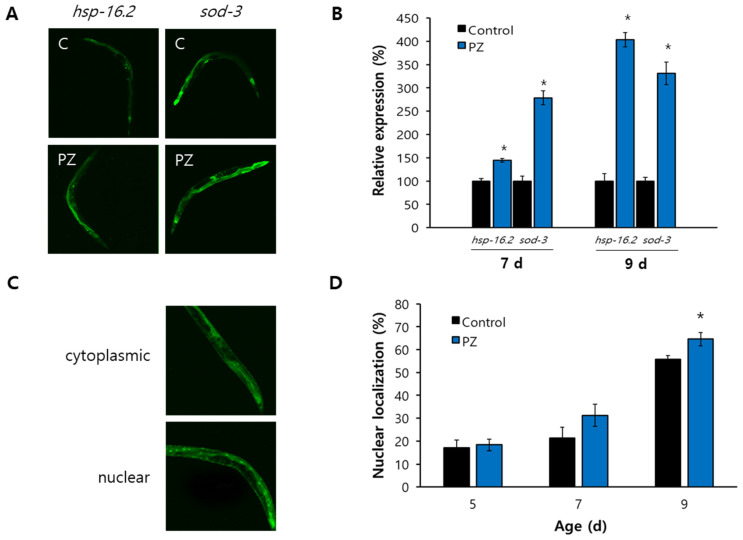
Transcriptional regulation of stress-responsive genes by phlorizin. (**A**) GFP expression induced by either *hsp-16.2* or *sod-3* promoter was observed in a fluorescence microscope. (**B**) Quantification of fluorescence was determined using a fluorescence multi-reader. Relative percent expression is shown compared to 100% for untreated control. (**C**) Subcellular localization of DAF-16 fused to GFP was determined under a fluorescence microscope. (**D**) Percent of worms showing nuclear localization was compared between untreated control and phlorizin-treated group. Error bar indicates standard error. C, untreated control; PZ, phlorizin (10 μM); *, a significant difference (*p* < 0.05) compared to untreated control.

**Figure 5 antioxidants-11-01996-f005:**
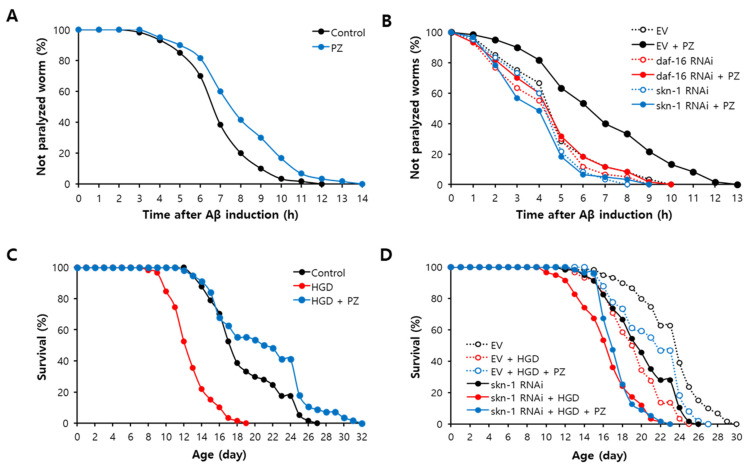
Effects of phlorizin on disease models of Alzheimer’s disease and diabetes mellitus. (**A**) Number of paralyzed worms after induction of Aβ transgene was counted every hour until all worms were paralyzed. (**B**) Effects of phlorizin on paralysis caused by Aβ were compared between control with EV RNAi and worms with *daf-16* or *skn-1* RNAi. (**C**) Survivals of worms were monitored for untreated control, HGD-treated group, and both HGD and phlorizin-treated group. (**D**) Role of SKN-1 in inhibitory effect of phlorizin on HGD-induced toxicity was examined using genetic knockdown of *skn-1*. PZ, phlorizin (10 μM); EV, empty vector; HGD, high-glucose diet (40 mM glucose).

**Figure 6 antioxidants-11-01996-f006:**
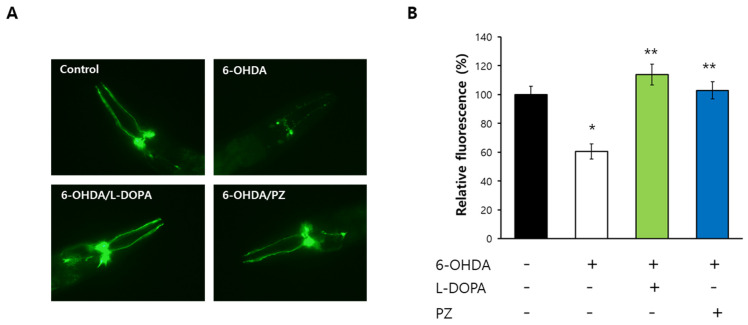
Inhibited degeneration of dopaminergic neurons by phlorizin. (**A**) GFP expressed in dopaminergic neurons was observed for untreated control, 6-OHDA-treated, 6-OHDA/L-DOPA-treated, and 6-OHDA/phlorizin-treated groups under a fluorescence microscope. (**B**) Quantification of fluorescence in the anterior head region was performed using image J program. Error bar indicates standard error. *, a significant difference (*p* < 0.05) compared to untreated control; **, a significant difference (*p* < 0.05) compared to 6-OHDA-treated group; 6-OHDA, 6-hydroxydopamine; L-DOPA, L-3,4-dihydroxyphenylalanine; PZ, phlorizin (10 μM).

**Figure 7 antioxidants-11-01996-f007:**
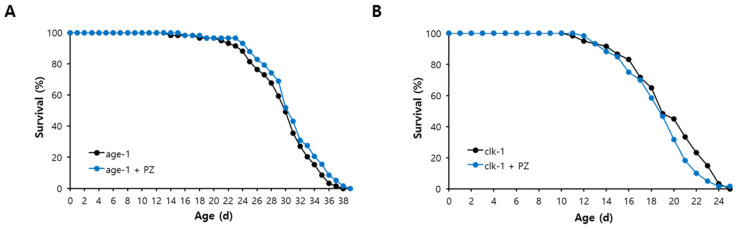
Underlying mechanisms involved in phlorizin-induced longevity. Lifespans of known long-lived mutants, *age-1* (**A**), *clk-1* (**B**), and *eat-2* (**C**), were monitored with or without supplementation with phlorizin. (**D**) Effect of phlorizin on lifespan was examined when expression of *daf-16* or *bec-1* was repressed by RNAi. PZ, phlorizin (10 μM); EV, empty vector.

**Figure 8 antioxidants-11-01996-f008:**
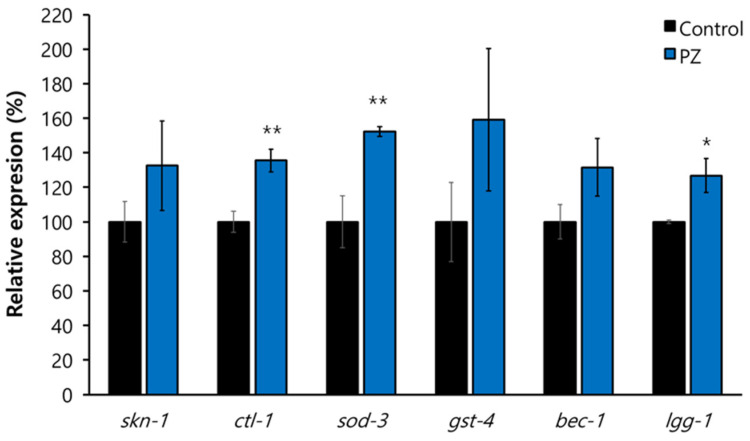
Effects of phlorizin on expression of anti-oxidant and autophagic genes. Quantitative RT-PCR was performed with total RNA extracted from 9-day-old worms treated with or without phlorizin. Relative percent expression of each gene in phlorizin-treated group was calculated compared to 100% for untreated control. *, 0.05 < *p* < 0.1 compared to untreated control; **, a significant difference (*p* < 0.05) compared to untreated control; PZ, phlorizin (10 μM).

## Data Availability

Not applicable.

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
