# Peer review of "Anti-Oxidant and Anti-Aging Effects of Phlorizin Are Mediated by DAF-16-Induced Stress Response and Autophagy in Caenorhabditis elegans"

_antioxidants, 2022, doi:10.3390/antiox11101996_

Round 1

Reviewer 1 Report

This is an interesting well written manuscript with experiments carefully designed and results well presented. While it has been known that flavonoids may promote survival or extend lifespan in some model organisms such as fruit flies, worms, and mice, this paper extended beyond what we have known by providing some insightful information concerning the underlying mechanisms. However, there are still some concerns/issues that may need to be addressed:

1. About Anti-oxidant effect of phlorizin: as it has been well established that many flavonoids are capable of scavenging free radicals or ROS, largely due to its multiple hydroxy groups that can react with free radicals. Surprisingly, this action was not observed in this study even though this molecule possess three hydroxy groups. Please discuss about this result.

2. About the use of RNAi: while the data show that the use of RNAi abolished the effects of phlorizin in several assays, no data confirm that the genes targeted by RNAi were ablated. In addition, since there are multiple isoforms of DAF-16. It is unclear which ones are targeted, or all of them.

3. About physiological relevance: While some flavonoids haven been shown to extend lifespan or survival of lower lever of organisms, such as flies, worm, and yeasts, but in rodents, the efficacy is either null or mild. One of the reasons could be due to the low bioavailability of flavonoids. Please comments on how the observed effective doses of this compound is physiologically relevant, given that the achievable blood levels of flavonoids in general after oral intake is lower than 1 uM in rodents and humans and quickly transformed in the liver.  

4. About autophage: the data may not convincingly show that phlorizin acts via this mechanism, as the data show that there are no significant differences in the expression levels of autophagic markers such as bec-1 and lgg-1 gene expression.

5. About DAF-16-mediated mechanism: Also this is doubtful as deletion of DAF-16 itself greatly reduced lifespan, so in this model, any lifespan-promoting agent could have lost its efficacy.   

6. About Introduction: This section could be shorten and more focused as it is written like a review paper.

Author Response

This is an interesting well written manuscript with experiments carefully designed and results well presented. While it has been known that flavonoids may promote survival or extend lifespan in some model organisms such as fruit flies, worms, and mice, this paper extended beyond what we have known by providing some insightful information concerning the underlying mechanisms. However, there are still some concerns/issues that may need to be addressed:

  1. About Anti-oxidant effect of phlorizin: as it has been well established that many flavonoids are capable of scavenging free radicals or ROS, largely due to its multiple hydroxy groups that can react with free radicals. Surprisingly, this action was not observed in this study even though this molecule possess three hydroxy groups. Please discuss about this result.

Generally, phytochemicals having hydroxy groups have ROS scavenging activity. However, the exact structure-activity relationship for ROS scavenging activity is still not clear. For example, luteolin, luteolin tetramethylether, and kaempherol, all of which have 4 hydroxy groups, showed 0, 2.67 and 3.57 times of Trolox equivalent antioxidant activity, respectively (Free Radic Biol Med. 1997;22(5):749-760). ROS scavenging activity of phytochemicals, including phlorizin, was shown in vitro at high doses that are hard to achieve in vivo. Previous studies suggested that in vivo antioxidant activity of luteolin, curcumin and resveratrol is due to the activation of stress-responsive transcription factor, Nrf2, and not related to direct ROS scavenging activity because of a poor bioavailability in vivo (J Nutr Biochem. 2014;25(6):581-591). We discussed this and the physiological relevance issues the reviewer’s raised in the “Discussion” section in the revised manuscript (Line 402-409).

  1. About the use of RNAi: while the data show that the use of RNAi abolished the effects of phlorizin in several assays, no data confirm that the genes targeted by RNAi were ablated. In addition, since there are multiple isoforms of DAF-16. It is unclear which ones are targeted, or all of them.

Feeding of RNAi clones from RNAi library to the wild-type laboratory strain N2 is a standard gene knockdown method. As the reviewer pointed out, not all RNAi clones show detectable phenotypes of the corresponding genes in RNAi library. We used three RNA clones in this study, daf-16, skn-1 and bec-1. Those RNAi clones were already used for gene knockdown in many studies and showed expected phenotypes with the feeding RNAi, the same method we used in this study. For example, regulation of lifespan by daf-16 RNAi (Nature. 2010;466(7305):498–502), change in stress response by skn-1 RNAi (PLoS One. 2021;16(7):e0249103) and decreased proteasome activity with bec-1 RNAi (Cells. 2020;9(8):1858) were reported using the same RNAi clones. For daf-16 RNAi, we used full-length daf-16 RNAi clone. We added “full-length” in the revised manuscript (Line 320).

  1. About physiological relevance: While some flavonoids haven been shown to extend lifespan or survival of lower lever of organisms, such as flies, worm, and yeasts, but in rodents, the efficacy is either null or mild. One of the reasons could be due to the low bioavailability of flavonoids. Please comments on how the observed effective doses of this compound is physiologically relevant, given that the achievable blood levels of flavonoids in general after oral intake is lower than 1 uM in rodents and humans and quickly transformed in the liver.

The reviewer pointed out a very important aspect of dietary interventions in lower organisms. Effective doses of antioxidants found in lower organisms are sometimes far beyond those applicable to higher organisms through dietary means. In addition, bioavailability of antioxidants is generally lower in higher organisms, including mammals. Therefore, it is very hard to predict physiological relevance or dose in mammals based on the findings with lower organisms. We addressed this limitation of this study and necessity of future study with mammals in “Discussion” section of the revised manuscript (Line 479-481).

  1. About autophage: the data may not convincingly show that phlorizin acts via this mechanism, as the data show that there are no significant differences in the expression levels of autophagic markers such as bec-1 and lgg-1 gene expression.

We agree the reviewer’s concerns regarding autophagy. The lifespan data showed knockdown of bec-1 abolished the longevity phenotype conferred by phlorizin, which strongly suggests the requirement of autophagy. A previous study also claimed that autophagy is involved in lifespan extension via reduced insulin/IGF-1-like signaling and dietary restriction using the same lifespan assay with bec-1 RNAi clone. Contrary to expectation, the expression of bec-1 was not changed and the p-value of lgg-1 expression was slightly higher than 0.05 (p = 0.052). Therefore, we are now performing the study focusing on effect of phlorizin on cellular autophagy using various genetic and biochemical tools and publish a following paper once we find an interesting results. We added this issue in “Discussion” section in the revised manuscript (Line 478-479).

  1. About DAF-16-mediated mechanism: Also this is doubtful as deletion of DAF-16 itself greatly reduced lifespan, so in this model, any lifespan-promoting agent could have lost its efficacy.

Our conclusions about DAF-16-mediated mechanism are based on several findings shown in this manuscript. Phlorizin enhanced nuclear localization of DAF-16 shown in Fig. 4. Downstream targets of DAF-16, sod-3 and hsp-16.2, exhibited increased expression by phlorizin in vivo (Fig. 4). Quantitative RT-PCR analysis also showed up-regulation of ctl-1 and sod-3, antioxidant genes regulated by DAF-16 (Fig. 8). Finally, the lifespan extension conferred by phlorizin was completely disappeared in age-1 mutant, in which lifespan increases via reduced insulin/IGF-1-like signaling and DAF-16 is required for lifespan extension (Fig. 7A). Based on those findings, we conclude the phlorizin-induced lifespan extension overlaps with the long lifespan by reduced insulin/IGF-1-like signaling. Additional result showing no lifespan extension by phlorizin in daf-16-knockdowned worms (Fig. 7D) further supported our conclusions.

  1. About Introduction: This section could be shorten and more focused as it is written like a review paper.

Following the reviewer’s suggestion, we shortened “Introduction” section.

Reviewer 2 Report

This study investigated antioxidant and anti-aging effects of a natural product phlorizin by using a Caenorhabditis elegans model. The manuscript comprehensively presents the current literatures concerning the subject of the aging-associated biological pathways and recent progress on using small molecules to reduce the aging effects. The authors tested anti-aging effects of phlorizin and investigated the molecular mechanisms.

The manuscript is, in principle, suitable for publication, just several minor suggestions.

1.     Besides the first time it appears in the main text, at all other places, Caenorhabditis elegans should be written as “C. elegans”.

2.     Figure 1: effects of phlorizin on the tested antioxidation assays are not dose dependent. Why is that?

3.     Line 223: “Mean survival time of the untreated control, 4.98 d.” This is not a complete sentence.

4.     Line 229: what is the p value between the 1 uM phorizin treated animals and the untreated control?

5.     Should the investigation of 10uM phorizin on cellular ROS level (figure 1D) be re-ordered to figure 1B? Also change the text order.

6.     Line 269: a space is missing before “78.0%”.

7.     Figure 3 caption: move the classification of 3 phases to the main text.

8.     Please add error bars to figures 2, 5 and 7.

9.     References: style as well as the paragraph format (font, size, line spacing etc) do not look correct. Please check.

10.  Check reference 35.

11.  Many references are very old. Can you replace them with more recent references?

Author Response

This study investigated antioxidant and anti-aging effects of a natural product phlorizin by using a Caenorhabditis elegans model. The manuscript comprehensively presents the current literatures concerning the subject of the aging-associated biological pathways and recent progress on using small molecules to reduce the aging effects. The authors tested anti-aging effects of phlorizin and investigated the molecular mechanisms.

The manuscript is, in principle, suitable for publication, just several minor suggestions.

  1. Besides the first time it appears in the main text, at all other places, Caenorhabditis elegans should be written as “C. elegans”.

Following the reviewer’s suggestion, we changed “Caenorhabditis elegans” from the second appearance to “C. elegans”.

  1. Figure 1: effects of phlorizin on the tested antioxidation assays are not dose dependent. Why is that?

There are several compounds showing rather bell-shaped, not-dose-dependent, bioactivities. For example, Cur2004-8, a curcumin derivative showed a significant antioxidant activity with 5 or 10 uM concentrations, but the effect was disappeared with higher concentrations (Drug Discov Ther. 2019; 13(4):198-206). Supplementation with phosphatidylserine also showed bell-shaped antioxidant activity; no antioxidant effect with 1 or 10 ug/L concentrations, a significant increase in resistance to oxidative stress with 100 ug/L concentration, and no change in response to oxidative stress with 1000 ug/L (Biogerontology. 2020 Apr;21(2):231-244). Interestingly, the effect of dietary restriction (DR), the most promising intervention retarding aging and extending lifespan, also exhibited bell-shaped effect on lifespan. The lifespan extending effect was not significant or little with a slight restriction and there is the optimum % restriction showing the biggest effect. However, the lifespan-extending effect of DR is disappeared with severe restriction (Toxicol Environ Health Sci. 2017; 9(1):59-63; J Nutr. 1986; 116(4):641-654). It is suggested that diet concentration has a bell-shaped relationship with lifespan, ranging from malnutrition, DR, to overfeeding, leading to nutritional toxicity (J Exp Biol. 2020; 223(Pt 23):jeb230185).

  1. Line 223: “Mean survival time of the untreated control, 4.98 d.” This is not a complete sentence.

We rewrite that sentence as “Mean survival time of the untreated control was 4.98 d” (Line 230).

  1. Line 229: what is the p value between the 1 uM phorizin treated animals and the untreated control?

P value of the comparison between 1 uM phlorizin and the untreated control was addressed in the revised manuscript (Line 223).

  1. Should the investigation of 10uM phorizin on cellular ROS level (figure 1D) be re-ordered to figure 1B? Also change the text order.

We re-ordered the Fig. 1D to Fig. 1B and changed the text order as the reviewer suggested.

  1. Line 269: a space is missing before “78.0%”.

It is corrected.

  1. Figure 3 caption: move the classification of 3 phases to the main text.

We moved the explanation about 3 phases to the main text of the revised manuscript (Line 272-276).

  1. Please add error bars to figures 2, 5 and 7.

During the lifespan assay, we employed the log-rank test for statistical analysis, which is one of most widely used statistical method for the comparison of lifespans between populations. The log-rank test compares survival curves of two different populations collected throughout the lifespan as a whole, not comparing percent survivals on each time point individually. Therefore, presenting the results from independent replicative experiments were preferred to adding error bars in a single survival curve as evidence for reproducibility of the observed result. We performed three independent replicative lifespan assays for each experiment and showed the results in Supplementary materials Table S2 and S6.

  1. References: style as well as the paragraph format (font, size, line spacing etc) do not look correct. Please check.

We changed the reference style following the journal’s guideline.

  1. Check reference 35.

Reference 35 was replaced with valid reference.

  1. Many references are very old. Can you replace them with more recent references?

Following the reviewer’s suggestion, we replaced some old references to recent ones.